# Technologies supporting vision screening: a protocol for a scoping review

Qasim Ali ,[1] Ilona Heldal ,[1] Carsten Gunnar Helgesen,[1] Gunta Krumina,[2] Marianne Nesbjørg Tvedt[1]

¹Department of Computer science, Electrical engineering and Mathematical sciences, Western Norway University of Applied Sciences, Bergen, Hordaland, Norway
²Department of Optometry and Vision Science, University of Latvia, Riga, Latvia

**Correspondence to**
Ilona Heldal; ilona.heldal@hvl.no

## ABSTRACT

**Introduction** Vision problems affect academic performance, social and mental health. Most traditional vision screening methods rely on human expert assessments based on a set of vision tests. As technology advances, new instruments and computerised tools are available for complementing vision screening. The scoping review based on this protocol aims to investigate current technologies for vision screening, what vision tests can be complemented by technologies, and how these can support vision screening by providing measurements.

**Methods and analysis** The planned review will utilise the PRISMA extension for Scoping Reviews (PRISMA-ScR) tool. Electronic search will be performed in databases, including Web of Science, MEDLINE (Ovid), Scopus, Engineering Village, Cochrane and Embase. We will perform a systematic search in selected reference databases without the limitation on publications dates, or country of studies. Reference management software, like EndNote and DistillerSR, will be used to remove duplicate entries. Two authors will independently analyse the studies for inclusion eligibility. Conflicts will be resolved by discussion. We will extract the types of technologies, types of vision tests they complement and the measurements for the included studies. Overall findings will be synthesised by thematic analysis and mapping to the logic model.

**Ethics and dissemination** Ethical approval is not required for this review, as it will only summarise existing published data. We will publish the findings in an open access, peer-reviewed journal. We expect that the review results will be useful for vision screening experts, developers, researchers, and policymakers.

## Strengths and limitations of this study

► Summarise published literature on up-to-date technologies supported vision screening with focus on school-aged children.
► Comprehensively examine vision screening studies using technologies with no time, or geographical restrictions.
► Provide evidence and reliability of using vision screening technologies.
► Not focusing on technologies for severe visual impairment or severe clinical cases.

overlapping standard tests, for example, visual acuity, colour vision, eye accommodation, ocular alignment, stereo vision and visual field.[5] Some of the tests are supported by technologies.

Traditional approaches are employed by professionals for vision screening of children.[6] Traditional vision screening methods rely on assessments from experts based on screening techniques are supporting vision tests. These techniques include picture tests (eg, tumbling 'E', Bailey–Lovie chart, Sloan letter (ETDRS) test, or Landolt's Broken Ring chart for distance visual acuity),[7 8] Ishihara test for colour vision[9] and stereo fly test for binocular vision disorders.[10] The methodologies to perform vision screening or diagnosis of vision problems require a high level of vision expertise, and a test battery.[11 12] These methods are resource demanding and challenging, especially for young children with poor collaboration abilities.[13]

Vision specialists (eye-care practitioners) are few, and do not have enough resources to screen all. This screening is time-demanding and resource-demanding and rarely includes the screening of the functional vision. As a result of ongoing research and development, many instrumental vision screeners between 'table-tops or handheld devices',

## INTRODUCTION

Early detection and effective treatment of vision problems are essential to reduce visual dysfunction in children. Studies argue that early diagnosis can prevent or minimise complications associated not only with vision impairment,[1 2] but later with poor academic performance, social, physical and mental health problems among children and adults.[3 4] Currently, vision specialists are responsible for testing visual functions and planning interventions. Several sets of vision screening batteries are available, including

computerised, web and mobile applications are available for functional vision screening.

Technology support can be achieved from specially designed instruments or computerised programmes. Such instruments are, for example, photoscreeners used to evaluate refractive errors, media opacities or eye misalignment.[14] Instrumental vision screening devices use state-of-the-art technologies, including autorefraction, retinal birefringence and photo screening techniques.[15] These devices are time efficient and provide comprehensive, objective measurements. As an example, Silverstein and McElhinny[16] showed that the average screening time of 120s for a child screened with the traditional method (optotype and stereoacuity) can be reduced to 30s with utilising a photoscreener. This is further corroborated by research showing that children with language skills or developing mental delay can be screened using instrumental vision screening methods.[7] Jesus et al[17] concluded that instrumental technology for objective refraction measurement could support subjective refraction techniques. A policy statement from the American Academy of Paediatrics, American Academy of Ophthalmology, American Association for Paediatric Ophthalmology and Strabismus and American Association of Certified Orthoptists have recommended the use of photoscreeners and handheld autorefractor devices as an alternative method for amblyopia and strabismus screening of children from 3 to 5 years old.[18] The instrumental vision screeners are categorised as 'table tops or handheld devices', for example, Plusoptix S12[19] and Spot Vision Screener[20] are handheld devices, while Zeiss Visuref[21] and iScreen 3000[22] are 'table-tops devices'

Computerised vision screening programmes help vision experts to accomplish a broad range of vision tests, including visual acuity, visual efficiency skills, colour vision, stereo vision, contrast sensitivity, visual field test or oculomotor behaviour of the eyes.[6 12 23] Usually, the programmes run on portable computers and often provide a self-assessment environment without necessary supervision.[24] Layperson, such as nurses or educators, can be trained to operate computerised vision testing systems. Visual Efficiency Rating is an example of a software programme designed for school nurses to assess accommodative, binocular and some oculomotor skills.[25] Due to the possibilities of eye-tracking technologies allowing to follow the left and right eyes separately, the oculomotor behaviour of eyes, for example, saccades, fixations and smooth pursuit movements, can be examined. Since portability and robustness are in particular focus, mobile and web technologies attract researchers and developer's attention. These computerised programmes aim to support vision screening time efficiently, with objective measurements, and can produce reliable and evidence-based data by laypersons.

Due to the variety and possibility of the available technologies, it is essential to investigate these technologies together to suggest appropriate and up-to-date support for broad vision screening of children and determine the state-of-the-art for further technology improvements. Some earlier reviews examined technologies for a part of a vision assessment battery. O'Hara and O'Hara[26] Nottingham et al[14] and Kaseem[15] examined the autorefractors, photoscreeners devices or optotype software. Kaseem et al[15] discussed a paediatric vision scanner device that uses the retinal birefringence technique. Furthermore, a scoping review by Yeung et al[27] explored the availability of web and mobile applications under eHealth tools.

The study described in this protocol will investigate a scale of broader technologies and assess their potential impact on vision assessment. To the best of our knowledge, there is no existing systematic literature review that considers all these technologies, and it will help the researchers to see the compiled information about vision screenings supported by technologies and to develop new ideas. Moreover, vision and other specialists will be able to read the review and choose the appropriate vision screening tool. This scoping review aims to include but is not limited to instrumental, computerised, mobile and web applications, eye-trackers, image processing, machine learning and computer vision approaches. All developed technologies will be investigated, and we will report the accuracy of such technologies if the studies provide them. The accuracy can be, for example, success rate, sensitivity, specificity and other factors influencing it, such as resolution, range, reliability and performance or error measures of the technologies.

Figure 1 shows a model guiding us to conceptualise how a range of technologies can be used for evaluating the different measurements, for example, fixation, saccades, smooth pursuit, pupil size or visual acuity. This scoping review aims to identify possible technologies for supporting screening for multiple problems. Therefore, we are not going to limit this scoping review to just instrumental and computerised technologies. Moreover, this review will highlight the intended stakeholders who could use the required technology for vision screening. A broader motivation behind the review is to identify the state-of-the-art technologies that can support vision screening at schools for all children by laypersons.

This paper outlines the protocol for a scoping review to summarise the available technologies, based on the type of vision test they can complement and the screening measurements the technologies can produce for complementation. Since instruments or computerised tests do not perform all vision tests, it is essential to observe the measurement parameters and their accuracy.

## Scoping review research questions

In order to determine the current state-of-the-art technologies and identify their support for vision screening, the scoping review will answer the following research questions:

1. What types of technologies are in use for vision screening?

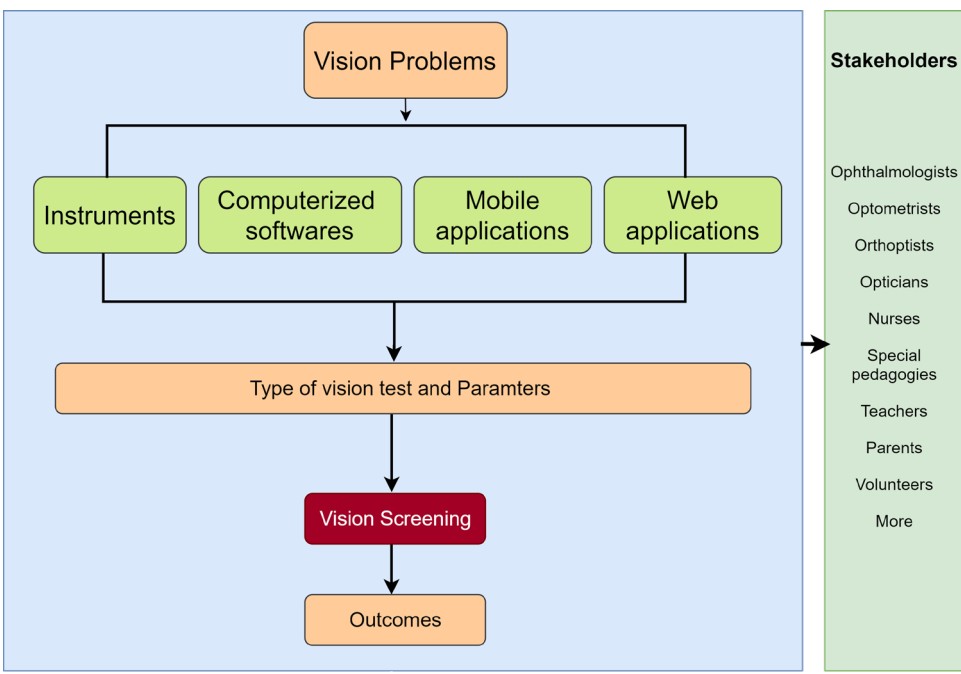

**Figure 1** Logic model represents the use of technologies for vision screening and associated stakeholders for vision screening. The scoping review will include technologies available for one or more vision problems, the measurement parameters used by that particular technology and the outcomes.

2. Which vision tests are complemented with these technologies?
3. Which vision function parameters and eye measurements can be performed by technologies comparing to manual vision screening?

## METHODS AND ANALYSIS
### Registration and protocol
The protocol is registered in the Open Science Framework (OSF) registry and is available to the public at https://osfio/u7m42/. We will update the protocol in the OSF registration if needed. This protocol is reported according to the relevant section of the Preferred Reporting Items for Systematic reviews and Meta-Analyses extension for Scoping Reviews (PRISMA-ScR) checklist[28] (online supplemental appendix 1).

### Inclusion criteria
#### Types of studies
All primary studies, systematic reviews, white papers and other reports will be included in the review. Technologies of interest include, but are not limited to, photoscreeners, autorefractors, computer software's, perimetry systems for the visual field, retinal birefringence, mobile, tablet and web applications. The authors of this protocol study are familiar with 'English', 'Norwegian', 'Urdu', 'Latvian', 'Swedish', 'Danish', 'German', 'Hungarian', 'Romanian' and 'Russian' languages. Therefore, the scoping review will include the literature written in the mentioned languages.

#### Types of participants
We are focusing on school-age children from 5 to 18 years old. We will exclude children with severe vision diseases, cognitive and mental disorders where the child clearly requires assistance. Therefore, only those studies will be included where children do not require special assistance from vision specialists for vision screening. Traditional and functional vision problems will be considered, and only those studies will be included that involved human participants without any limitation on sample size. There will be no limitation for any ethnic group or country.

#### Types of outcome measures
We will include studies that report the eye measurements recorded by technology because every visual function test has different measurement parameters.

### Search strategy and information sources
A librarian-assisted search will be conducted in the following databases:
1. Web of Science.
2. Scopus.
3. Engineering Village (Compendex).
4. Cochrane CENTRAL (which contains the Cochrane Eyes and Vision Trials Register).
5. MEDLINE (Ovid).
6. Embase.

The search strategy will combine keywords and subject headings (where thesauruses are available), using search terms for vision screening combined with search terms for instrumental technologies, for example, 'handheld', 'table-top', to computerised, for

example, 'software', 'eye-tracking', 'smartphone', 'web' and 'machine learning' and other devices such as 'head-mounted displays' and 'tablet'. The search strategy of MEDLINE (Ovid) is included in the online supplemental appendix 2. The reference list of the included papers will be checked to identify any additional studies not retrieved by the database searches. We will also search for articles citing the included papers, using Scopus. The next step will be searching for grey literature, using electronic sources such as Google, clinicaltrial.gov, euscreen.org. A complete list of resources is included in the online supplemental appendix 3.

## Study selection

All search results will be retrieved and imported to reference management software (EndNote library) and we will remove duplicate entries. Later, all studies will be uploaded to a review management software Rayyan (https://rayyan.qcri.org/) or DistillerSR (https://www.evidencepartners.com/products/distillersr-systematic-review-software) and assess all articles, by title and/or abstract, to identify the studies that meet the inclusion criteria. Full-text screening of the selected articles will be completed for eligibility. Two independent reviewers will conduct the eligibility assessment and conflicts will be resolved by discussion with the third reviewer. We will compile a PRISMA-ScR flow diagram to summarise the screening process of the study.

## Data charting process

Data charting forms will be developed in Google Forms or DistillerSR based on the data items described below. Before using these forms, all reviewers will test them on three studies. In the screening process, two reviewers will perform the data charting of including studies independently. Because of the broad scope of the included studies, the data charting process will be iterative, and amendments will be made as required. A discussion with the third reviewer will resolve any differences in charting. In case of vague or incomplete information, we will contact the study authors via email for up to three attempts.

## Data items

We will extract the following data items during the charting process.
1. Publication characteristics:
   – Title.
   – Year of publication.
   – Study design.
   – Country of origin.
   – Age group of the study population.
   – Study setting.
2. Study details:
   – Name of technology (instrumental, computerised, mobile or web).
   – Type of measurement.
   – Intervention types (static or animated).

– Machine learning or image processing technique used (if applicable).
3. Stakeholders:
   – Healthcare professionals.
   – Teachers.
   – Layperson.
4. Association with the test:
   – Technologies suitable for the screening of vision problems.
   – Data, measurements used within the tests.
5. Evaluated outcomes of the study:
   – The precision of measurements' acquisition.
   – Factors influencing the accuracy of the measurements with technologies, for example, resolution and range success rate, sensitivity and specificity.
6. Authors' reflections:
   – Suggestions and future directions of the authors.

## Synthesis of results

From the data charting, we will synthesise the results by mapping the extracted evidence to our logic model shown in figure 1. Data of each study will be examined using the pathway of the logic model. We will summarise the classification of technologies with the associated measurements, stakeholders who can use the technology, vision screening, and outcomes.

## Patient and public involvement statement

This review will be performed without specific patient or public involvement. It is based only on existing published literature.

## ETHICS AND DISSEMINATION

Ethical approval is not required for this scoping review, as it will only summarise the existing published data. We will publish our findings in open access, peer-reviewed journal. A summary of the results will be generated for website posting, and stakeholder meetings. We expect that the findings will be useful for the stakeholders involved in vision screening: ophthalmologists, optometrists, orthoptists or special education teachers[29] and give valuable insights for technology developers. We also anticipate that the results will advocate for creating comprehensive vision screening policies for the children who need eye care. Furthermore, the data will provide valuable information to the researchers for future research to investigate the identified technologies.

**Contributors** QA is the primary author responsible for this document with IH's and CGH's supervision. IH conceived the idea for the review. QA designed and drafted the protocol paper with IH, CGH, GK and MNT suggestions. QA, IH and CGH are researchers from engineering and information systems. MNT is a researcher in library science who contributed to improving search strategies and finding relevant databases. GK is a domain expert from optometry who critically reviewed the protocol and she was included in several iterations when vision expertise was necessary.

**Funding** This research is financed by the Research Council of Norway, project no: 267524/H30.

**Competing interests**  None declared.

**Patient consent for publication**  Not required.

**Provenance and peer review**  Not commissioned; externally peer reviewed.

**Open access**  This is an open access article distributed in accordance with the Creative Commons Attribution 4.0 Unported (CC BY 4.0) license, which permits others to copy, redistribute, remix, transform and build upon this work for any purpose, provided the original work is properly cited, a link to the licence is given, and indication of whether changes were made. See: https://creativecommons.org/licenses/by/4.0/.

**ORCID iDs**
Qasim Ali http://orcid.org/0000-0003-3314-3086
Ilona Heldal http://orcid.org/0000-0003-1149-8820

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
