## [Reviewer comments · BMJ Open]

ARTICLE DETAILS

TITLE (PROVISIONAL)	Technologies supporting vision screening: a protocol for a scoping review
AUTHORS	Ali, Qasim; Heldal, Ilona; Helgesen, Carsten Gunnar; Krumina, Gunta; Tvedt, Marianne Nesbjørg

VERSION 1 – REVIEW

REVIEWER	Moore, Bruce New England College of Optometry
REVIEW RETURNED	13-Mar-2021

GENERAL COMMENTS	Clearly, this is a “very different” review of a paper that was submitted to a journal for my review, at least in my own personal experience. I do understand the importance of clinical reviews, guidelines, etc. but have not had a prior review experience with this particular type of submission. It is frankly, in the case of vision screening technologies, a difficult subject to bring clarity to, and I wish the authors and participants of this project the best of luck. Basically, I have no issue with the bulk of what is presented here, which follows the established author guidelines from BMJ Open, etc. Nonetheless I will take this opportunity to voice some comments and suggestions to the authors about their upcoming and difficult task. These suggestions can be considered if the team feels them useful, or ignored if they so choose, but they are points about this subject that I have in my own mind and experience. So here goes... Line 17: I do not think Metsing’s classification is useful, as the three items continually merge and shift in the real world; how do you really separate items 2 and 3? Well, you can’t. Furthermore, just about every single new vision screening technology out there nowadays has a computer of some type built in, so how can there be a separate category, and in the future there will be even more merging of online and “in the room” screening by the same technologies. How the result gets read and interpreted may be either online or present, and this already exists today. Another example is VA screening in the room or on a video-screen – how different is that in concept and in reality? Yes, implementation, but of course that is the basis of measuring or screening for VA anyhow. So, I think that classification is meaningless and ought to be avoided. My own view, again take it or leave it, is that you can categorize most everything existing today as fitting into 3 categories, refractive error, visual acuity, and binocularity however you choose to define that, or alternatively, structural (VA and probably Blinq) and functional (VA
--

and binocularity). At least that is the way many of us in the US thinking about these issues view the milieu. Again, take it or leave it, but please leave Metsing.

Line 25: "Traditional" screening techniques can be complicated or simple in their approach and design, with simple for non-professionals to carry out within a tight protocol totally centrally important for good results. These techniques can be set up either in a simple or complex manner, with results generally following the ability of screeners to follow protocol. For example, very simplified VA screening charts have been designed to make the presentation to even a young child pretty "idiot proof", with higher testability rates as determined in studies than a supposedly simple instrument based technique that purports speed and efficiency. Examples are single optotype flip books with crowding and an answer sheet designed to keep things simple. Just because some of the newer high tech approaches appear simple, does not mean that they actually are. An example would be the highly variable rates of testability between the now several photoscreening devices currently being used, with variability based on lighting, reflectance, attention, etc. Different devices have very different testability in real life that the manufacturers just gloss over.

Line 54: The type of technologies described here that go beyond refraction are a very broad category, but one must be vigilant to not be overly impressed by simply adding more and more separate procedures to a single instrument to make it "more comprehensive". Not every one of those add-ons may be useful or even minimally accurate in what they purport to do. Just adding more accessories does not necessarily make a smooth running vehicle (BMW anyone?) better, nor does it with vision screening equipment. Obviously, more can be more expensive, more complicated, and more prone to ultimate failure. Also sometimes, pretty useless, just the proverbial "bells and whistles". Having a device do, say, six things, does not necessarily make a better screening process. Simplicity can be a virtue.

No line in particular: The underlying concept of this project (I assume) is to determine what actually works to improve our ability to screen the vision of children of varying ages and situations. The authors will necessarily throw the net out widely to identify multitudes of papers to base their analysis and thence their conclusions on. Here lies the biggest and most unavoidable problem. They need to determine the quality of the extant work on the subject within the literature. Unfortunately I know from experience that the literature is extremely problematic on the subject of instrument based vision screening, however one defines it. There are many studies published even in high quality journals that suffer from problems including poor study design, small sample size, selected tertiary care clinical sites that are not representative of broad national populations, and particularly the reality that most studies have a predominance of subjects clustered around essentially "normal" refractive errors (say -2.00 to +2.00D). That

	means few with higher refractive errors (myopia, hyperopia, and astigmatism all), but nonetheless producing regression curves arbitrarily drawn way beyond any semblance of rationality. Yes, you can draw the line out as far as you choose, but the accuracy of these higher refractive error interpretations are often very wrong as the preponderance of the data come from the middle, not the ends of the curve. I am sure that the authors are aware of this, but it is a serious problem, as these are the kids that we need most to successfully detect, but the built in algorithms in the devices generally do a poor job of accurately dealing with. The authors of this project will really need to work very hard to cut through that fog. In particular, the literature for the current and previous generations of photoscreeners (i.e. Spot, PlusOptix, and 2Win back to their proverbial grandfather, the Photorefractor 2) have few moderate to high hyperopes and the limited data frequently presented shows far greater variability of measure beyond about +3.50D or so, with an actual inability to gain a valid measure of many of those eyes, missing one of the greatest risks of amblyopia. To some degree the same for anisometropia and high myopia as well. However the papers often gloss over that with a small handful of data points that are pretty much blown off by continuing the regression curve to the end of the page. Thus, the quality of many papers is nowhere as good as some claim. Basing a major examination and assessment of instrument based screening and not having the ability of filtering that noise out is going to be a major task for this study group. I do hope they find a way to overcome what I fear is going to be a big problem. I think it is critical that they have team members who are extremely familiar with the technologies, have considerable clinical experience, and are highly selective in which papers / studies they will include for analysis. One other suggestion, take it or leave it as I said above; do not ignore the so-called traditional approaches to screening. Some of the new devices attempt to add on some or a lot of their versions of VA, stereo, visual skills, etc. Sometimes the older ways have advantages, not the least is a cost structure a fraction of what the newer devices cost, and with results that may not even be as good, or importantly, as flexible for a child that is more difficult. In summary, this is going to be a difficult task for the authors to do justice on. My suggestions and comments may be useful, or not. Good luck. I eagerly await their results.
--	--

REVIEWER	McGilton, Katherine Toronto Rehabilitation Institute
REVIEW RETURNED	20-Mar-2021

GENERAL COMMENTS	This protocol focused on identifying the current state of the art technologies supporting vision screening. It will investigate what technologies are available to complement vision screening tests and the reliability of the results. Overall the review is clear and the logic model serves as a good guide for the review.
---

	Please consider the following queries to further enhance the clarity of the protocol  1) What is the target group for these technologies? Any age group or children? For example vision screening measures may be different for persons with dementia vs children vs young adults. I note that 'infants' are excluded. 2) How are technologies defined? 3) In the logic model there is a section including 'associated stakeholders'. Patient/Society participation in research is becoming more and more important for research projects. How do the researchers plan to include stakeholders during their review? 4) Will this review focus on screening or assessment tools for vision that will be investigated? Who would be completing the tools and should this also be captured so that the findings are useful for practice 5) What methodology is guiding the review process? 6) Accuracy will be described for each test. How is accuracy defined? 7) Will a librarian scientist be available to assist with the search? 8) There are no language restrictions. This is great but how is this feasible? 9) How will the grey literature search results be analyzed? 10) How will this review help to determine "their usefulness for vision screening" 11) What was learned in the other reviews? How does this review build on that work?
--	---

VERSION 1 – AUTHOR RESPONSE

Reviewer 1 comments (Dr. Bruce Moore, New England College of Optometry)	Answers
Line 17: I do not think Metsing's classification is useful, as the three items continually merge and shift in the real world; how do you really separate items 2 and 3? Well, you can't. Furthermore, just about every single new vision screening technology out there nowadays has a computer of some type built in, so how can there be a separate category, and in the future there will be even more merging of online and "in the room" screening by the same technologies. How the result gets read and interpreted may be either online or present, and this already exists today. Another example is VA screening in the room or on a video-screen – how different is that in concept and in reality? Yes, implementation, but of course that is the basis of measuring or screening for VA anyhow. So, I think that classification is meaningless and ought to be avoided. My own view, again take it or leave it, is that you can categorize most everything existing today as fitting into 3 categories, refractive error, visual acuity, and binocularity however you choose to define that, or	Thank you for the great comments and suggestions. We want to divide manual or traditional and instrumental screening and technologies based computerized. In manual or traditional screening, a person (often an expert) examining the eye and discussed participants are involved. Manual screenings usually include limited VA, stereovision and colour vision plates. As we describe in the protocol, we do not consider these tools instruments. These depend on the test battery and help human (often) experts to screen or measure. Instrument or computer-based technologies measure VA and include different binocular vision tests, color vision tests, simple visual field tests, or are used for the assessment of eye refraction by photorefractive principle. Instruments are more dedicated; they often can perform a limited number of tests for screening

alternatively, structural (VA and probably Blinq) and functional (VA and binocularity). At least that is the way many of us in the US thinking about these issues view the milieu. Again, take it or leave it, but please leave Metsing.

vision impairments, e.g., a photo screener. It is difficult to update them or complement them with other test batteries, calculations. Computer-based technologies include programs with tests for vision screening besides the several different other aims computers have. They can be extended, used together with other programs or in distributed settings. Accessibility of instruments differs from computer-based applications. Instruments and computer-based technologies will tell us different ways to measure something.

Metsing used three categories. But we see instrumental and computerized are almost the same thing with slightest differences, as we described above. Many computerized technologies exist, such as computer software for eye-tracking, and measuring VA using a computer screen operated by software. Similarly, due to COVID19, we believe some ophthalmologists are working on teleophthalmology as well. However, we removed Metsing's classification and will focus on traditional and computer-supported screening (including instruments).

We will also classify vision problems and technologies used for the screening after completing our initial search.

We want to create a table to represent the vision problem and available technologies for it. To begin with the vision impairment and find possible technologies or technologies to see how they can be suitable support is a chicken and the egg problem. We think we need to investigate the many searches (a huge job) to see possibilities. But vision is more important. Therefore, thank you for the suggestion for the 3 categories: refractive error, visual acuity, and binocularity. If we begin with the categories, we can find (hopefully find) the same literature that should be included in several or all the three categories, but with measures to take away duplicate, this categorization is promising.

The overall aim is to support screening for many (for all school-aged children in the following step). This means that we would like to define good enough technologies that help non-professionals identify "possible" problems and provide systematic evidence to send the children with

Line 25: "Traditional" screening techniques can be complicated or simple in their approach and design, with simple for non- professionals to carry out within a tight protocol totally centrally important for good results. These techniques can be set up either in a simple or complex manner, with results generally following the ability of screeners to follow protocol. For example, very simplified VA screening charts have been designed to make the presentation to even a young child pretty "idiot proof", with higher testability rates as determined in studies than a supposedly simple instrument based technique that purports speed and efficiency. Examples are single optotype flip books with crowding and an answer sheet designed to keep things simple. Just because some of the newer high tech approaches appear simple, does not mean that they actually are. An example would be the highly variable rates of testability between the now several photoscreening devices currently being used, with variability based on lighting, reflectance, attention, etc. Different devices have very different testability in real life that the manufacturers just gloss over.	"possible vision impairment" to professionals. Our approach focused on overall vision screening. It is not limited to Visual acuity. We need to check children's binocular balance and accommodation ability. VA only works in one direction. It cannot tell us about accommodation or other ocular movements. We need to find additional methods how to check vision functions faster and for all school beginners. There are many instrument-based technologies available in the market and in the research process as well. As you pointed out, these devices can produce different results, and their testability varies. The differences may enhance limitations and may set additional requirements for acceptability. This review paper will find out the differences, reliability, and stakeholders of such devices. There are some handheld devices which can be used by volunteers, and the interpretation and results of the test are simplified by the devices. A computer today can be seen (or extended) as a collection of sensors that also can inform, be based on lighting, reflectance etc.
Line 54: The type of technologies described here that go beyond refraction are a very broad category, but one must be vigilant to not be overly impressed by simply adding more and more separate procedures to a single instrument to make it "more comprehensive". Not every one of those add-ons may be useful or even minimally accurate in what they purport to do. Just adding more accessories does not necessarily make a smooth running vehicle (BMW anyone?) better, nor does it with vision screening equipment. Obviously, more can be more expensive, more complicated, and more prone to ultimate failure. Also sometimes, pretty useless, just the proverbial "bells and whistles". Having a device do, say, six things, does not necessarily make a better screening process. Simplicity can be a virtue.	We agree that simplicity, together with affordability, is a virtue in this case. To simplify, we need to know the main direction and make it affordable, and we need to know the main issues about current possibilities. -Identify different types of technologies and their effectiveness and success rate to give an overview which instrument is suitable or feasible to diagnose the specific type of vision problem or target people. -The main aim of scoping review is to summarize or find used and experienced as useful technologies. E.g. for binocular vision screening, we will see which type of technologies or instruments are available. What kind of functions and technologies do these devices/instruments provide -It will be helpful for software developers, professionals, volunteers, teachers, and policy-makers. The developers met the wall today. They can add a new/changed functionality (e.g. by using virtual reality for some tests or

	"computerizing" or gamifying some other test), volunteer screening can be improved, but teachers or policy-makers do not necessarily know about problems. The Norwegian policy-makers are still waiting for evidence from RCT-s to presenting systematic data about the value of the data. But, this can hardly be designed/collected in Norway due to the missing experts. But, technologies cannot be developed if we don't know what they already (or can) support and how today, based on a more overall picture about vision impairments. -This work is a step on the road, and we know that the road has a number of obstacles.
No line in particular: The underlying concept of this project (I assume) is to determine what actually works to improve our ability to screen the vision of children of varying ages and situations. The authors will necessarily throw the net out widely to identify multitudes of papers to base their analysis and thence their conclusions on. Here lies the biggest and most unavoidable problem. They need to determine the quality of the extant work on the subject within the literature. Unfortunately I know from experience that the literature is extremely problematic on the subject of instrument based vision screening, however one defines it. There are many studies published even in high quality journals that suffer from problems including poor study design, small sample size, selected tertiary care clinical sites that are not representative of broad national populations, and particularly the reality that most studies have a predominance of subjects clustered around essentially "normal" refractive errors (say - 2.00 to +2.00D). That means few with higher refractive errors (myopia, hyperopia, and astigmatism all), but nonetheless producing regression curves arbitrarily drawn way beyond any semblance of rationality. Yes, you can draw the line out as far as you choose, but the accuracy of these higher refractive error interpretations are often very wrong as the preponderance of the data come from the middle, not the ends of the curve. I am sure that the authors are aware of this, but it is a serious problem, as these are the kids that we need most to successfully detect, but the built in algorithms in the devices generally do a poor job of accurately dealing with. The authors of this project will really need to work very hard to cut	Yes, the underlying motivation is correct. And yes, we are aware of throwing out a net and the biggest problem, to identify quality results. An author here is an expert in optometry, and we have connections with experts in special educations to discuss the quality of certain papers. Yes, we are going to meet studies with "poor study design, small sample size, selected tertiary care clinical sites that are not representative". Unfortunately, these studies are there, and misinterpretations can be made by many. If we wrongly identify the low-quality work due to our limited expertise (and high wish to improve technology support), we hope that enhancing these examples meet discussion and critiques from the reviewers and the (vision) professionals' community. One of the main motivations behind this work is to identify quality criteria for screening tests and the methodology the studies included here. We understand the fog behind existing tests, technologies and methodologies. One motivation to include grey literature from large international studies is to understand (possible) methodologies better, a bigger pattern.

through that fog. In particular, the literature for the current and previous generations of photoscreeners (i.e. Spot, PlusOptix, and 2Win back to their proverbial grandfather, the Photorefractor 2) have few moderate to high hyperopes and the limited data frequently presented shows far greater variability of measure beyond about +3.50D or so, with an actual inability to gain a valid measure of many of those eyes, missing one of the greatest risks of amblyopia. To some degree the same for anisometropia and high myopia as well. However the papers often gloss over that with a small handful of data points that are pretty much blown off by continuing the regression curve to the end of the page. Thus, the quality of many papers is nowhere as good as some claim.	
Basing a major examination and assessment of instrument based screening and not having the ability of filtering that noise out is going to be a major task for this study group. I do hope they find a way to overcome what I fear is going to be a big problem. I think it is critical that they have team members who are extremely familiar with the technologies, have considerable clinical experience, and are highly selective in which papers / studies they will include for analysis. One other suggestion, take it or leave it as I said above⁵. Some of the new devices attempt to add on some or a lot of their versions of VA, stereo, visual skills, etc. Sometimes the older ways have advantages, not the least is a cost structure a fraction of what the newer devices cost, and with results that may not even be as good, or importantly, as flexible for a child that is more difficult. In summary, this is going to be a difficult task for the authors to do justice on. My suggestions and comments may be useful, or not. Good luck. I eagerly await their results.	We believe in traditional ways of screening, but we also believe that this way is not enough for screening many children due to the resource and time limitations. Our fear is the same, to not be able to define a basic line for vision impairments that technologies can support, and we know that sometimes technologies (technology developers and providers) promise too much. In addition, we have technology providers in the background loudly arguing for their best solutions. There are also researchers like us, eager to better understand technology development for recognizing vision impairment. But, without this study, we cannot go further. We can only improve a test or an instrument with a (or a few) function(s). There is a mess outside with available technologies, and as you wrote earlier, we have many papers with questionable quality. Sometimes we (from the technology developer part) feel eager to investigate the situation. This also requires time and resources from many collaborators. I think our fear is not to not being able to see through the fog, filter the noise but not to try.

Reviewer 2 comments (Dr. Katherine McGilton, Toronto Rehabilitation Institute)	Answers
What is the target group for these technologies? Any age group or children? For example, vision	We will exclude children with severe vision diseases, cognitive and mental disorders

screening measures may be different for persons with dementia vs children vs young adults. I note that 'infants' are excluded.	where the child requires assistance. These children need to check vision by vision specialists who can adapt the vision tests according to the child's capability and understanding. Children with some mental disorders can do the screening without specific assistance. We updated this information in "Types of participants" section in this protocol.
How are technologies defined?	Vision specialists perform vision screening by manual approaches or instrumental devices and computerized supported programs. Manual test performed by professionals and defined test batteries for each test will be used. However, instrumental and laypersons can use computerized programs too, and one computer program or instrument can be used for more than one vision test.
In the logic model there is a section including 'associated stakeholders'. Patient/Society participation in research is becoming more and more important for research projects. How do the researchers plan to include stakeholders during their review?	We include research performed by vision specialists or educators. However, we have seen studies including volunteers and nurses. They can also conduct vision screening on a large population. In this scoping review process, we will extract the information of stakeholders provided in the articles.
Will this review focus on screening or assessment tools for vision that will be investigated? Who would be completing the tools and should this also be captured so that the findings are useful for practice	Screening is a battery that assesses the children who have one or two vision problems. Screening is a methodology to define that there is a vision. Assessment tools are manual tools that are used in practice. Assessments tools are more of diagnostic tools. We are focusing only on the screening part. The answer to the second question is that we will extract the information of users who uses the tools (stakeholders). They can be professionals such as nurses, optometrists, and non-professionals e.g. volunteers.
What methodology is guiding the review process?	This is a scoping review, and we are following Preferred Reporting Items for Systematic reviews and Meta-Analyses extension for Scoping Reviews (PRISMA-ScR). We have mentioned it in "3.1 Registration and Protocol" section of the paper.
Accuracy will be described for each test. How is accuracy defined?	We are interested in instruments and devices that used for vision screening. Literature gives

	information of such instruments or devices in terms of success rate, sensitivity, and specificity. For example, (ref3, Peterseim) used Spot Vision Screener in school- age children. They mentioned the success rate, sensitivity, and specificity of the device. This is how we will extract the data from articles. If the article publishes the data, we will define it. Otherwise, we will not mention that this accuracy is not defined.
Will a librarian scientist be available to assist with the search?	Marianne Nesbjørg Tvedt, one of the co-author of this protocol, is a librarian scientist. She helped us in defining search terms, and she will help us in electronic searches.
There are no language restrictions. This is great but how is this feasible?	Most abstracts are given in the English language. We can determine the inclusion of the article by reading the abstract, and later we will translate the article into the English language to extract the information. Further, people working on this project are multicultural. The authors speak/understand "English", "Norwegian", "Urdu", "Latvian", "Swedish", "German", "Hungarian", "Romanian" and "Russian" languages.
How will the grey literature search results be analyzed?	Government institutes or projects funded by European Unions publish results on corresponding websites. EUScreen¹, National health institutes of Norway,² are an example of grey literature. We will extract the information provided by grey literature in a separate section in our scoping review article. The included grey literature is mainly about screening larger populations, performing national screening programs. We are primarily interested in its methodologies and how they define accuracy for multi-national or larger screening projects.
How will this review help to determine "their usefulness for vision screening"	Many factors will determine the usefulness of technologies for vision screening such as, number of published articles, use of technology in clinical and non-clinical settings, the success rate of technologies, and current policies and legislation for the use of technologies. Helpful for readers to see the types of technologies available for vision screening.
What was learned in the other reviews? How	-To our knowledge, there is no such review

does this review build on that work?	paper exist which cover all type of technologies mentioned in our model diagram. However, some scoping review papers focused on a particular one or two types of technologies, such as Yeung et al(ref 1) focused on Mobile and Web apps as an e-Health tool for visual acuity. -A protocol published in 2019 (ref 2) focused only on smartphone technologies for detecting retinopathy. -Nottingham et al. (ref 4) gives an overview of the available instrumental and computerized technologies for vision screening in clinical and non- clinical settings. Kassem et al (ref 5) give a detailed overview of auto- refractors and photo screeners. -From the literature, we learned about the types of technologies available for vision screening and target population. We have seen advancements in technology assisting vision specialists in objective and even subjective measurements with greater success rates.
---	---